# Prognostic Impact and Clinical Implications of Unfavorable Upgrading in Low-Risk Prostate Cancer after Robot-Assisted Radical Prostatectomy: Results of a Single Tertiary Referral Center

**DOI:** 10.3390/cancers14246055

**Published:** 2022-12-09

**Authors:** Antonio Benito Porcaro, Andrea Panunzio, Alberto Bianchi, Marco Sebben, Sebastian Gallina, Mario De Michele, Rossella Orlando, Emanuele Serafin, Giovanni Mazzucato, Stefano Vidiri, Damiano D’Aietti, Alessandro Princiotta, Francesca Montanaro, Giulia Marafioti Patuzzo, Vincenzo De Marco, Matteo Brunelli, Vincenzo Pagliarulo, Maria Angela Cerruto, Alessandro Tafuri, Alessandro Antonelli

**Affiliations:** 1Department of Urology, University of Verona, Azienda Ospedaliera Universitaria Integrata of Verona, 37126 Verona, Italy; 2Department of Urology, IRCCS Ospedale Sacro Cuore Don Calabria, 37024 Negrar, Italy; 3Department of Pathology, University of Verona, Azienda Ospedaliera Universitaria Integrata of Verona, 37126 Verona, Italy; 4Department of Urology, Vito Fazzi Hospital, 73100 Lecce, Italy

**Keywords:** ISUP 1 prostate cancer, tumor upgrading, adverse pathology, prostate cancer progression

## Abstract

**Simple Summary:**

Unfavorable pathology in low-risk prostate cancer (PCa) is one of the most controversial subjects for the implications in clinical decision making when counseling patients. In the present study, we tested the hypothesis that unfavorable tumor upgrading, which we defined as International Society of Urological Pathology (ISUP) tumor grade groups greater than 2, might have a different prognostic impact when compared with no (ISUP 1) or favorable (ISUP 2) upgrading. Study findings show that low-risk patients with unfavorable tumor upgrading were more likely to have disease relapse, which occurred in 8.4% of cases and was associated with older age, PSA density ≥ 0.15 ng/mL/cc, and tumors being larger and extending beyond the gland. Unfavorable tumor upgrading is an issue to consider when counseling low-risk patients in order to avoid delayed treatments, which may impair cancer-specific survival. These findings represent a novelty for either urologists or radiation oncologists when counseling low-risk PCa patients.

**Abstract:**

Objective: to evaluate predictors and the prognostic impact of favorable vs. unfavorable tumor upgrading among low-risk prostate cancer (LR PCa) patients treated with robot-assisted radical prostatectomy (RARP). Methods: From January 2013 to October 2020, LR PCa patients treated with RARP at our institution were identified. Unfavorable tumor upgrading was defined as the presence of an International Society of Urological Pathology (ISUP) grade group at final pathology > 2. Disease relapse was coded as biochemical recurrence and/or local recurrence and/or presence of distant metastases. Regression analyses tested the association between clinical and pathological features and the risk of unfavorable tumor upgrading and disease relapse. Results: Of the 237 total LR PCa patients, 60 (25.3%) harbored unfavorable tumor upgrading. Disease relapse occurred in 20 (8.4%) patients. Unfavorable upgrading represented an independent predictor of disease relapse, even after adjustment for other clinical and pathological variables. Conversely, favorable tumor upgrading did not show any statistically significant association with PCa relapse. Unfavorable tumor upgrading was associated with tumors being larger (OR: 1.03; *p* = 0.031), tumors extending beyond the gland (OR: 8.54, *p* < 0.001), age (OR: 1.07, *p* = 0.009), and PSA density (PSAD) ≥ 0.15 ng/mL/cc (OR: 1.07, *p* = 0.009). Conclusions: LR PCa patients with unfavorable upgrading at final pathology were more likely to be older, to have PSAD ≥ 0.15 ng/mL/cc, and to experience disease relapse. Unfavorable tumor upgrading is an issue to consider when counseling these patients to avoid delayed treatments, which may impair cancer-specific survival.

## 1. Introduction

Prostate cancer (PCa) represents the second most diagnosed cancer in men worldwide [1] and has become an epidemic issue in the aging male population due to uncontrolled opportunistic screening [2,3]. Although most PCa cases are diagnosed at localized early stages, prognosis varies according to clinical risk categories that are not equivalent for the two main classification systems: the European Association of Urology (EAU) and the National Cancer Comprehensive Network (NCCN) [2,3]. Features defining the low-risk (LR) category are the same for the EAU and the NCCN. However, the NCCN also considers a subset of patients who are classified as very low-risk in the presence of a prostate-specific antigen (PSA) density (PSAD) < 0.15 ng/mL/cc and less than three biopsy-positive cores with cancer involving no more than 50% of each invaded core [3]. Accordingly, clinical decision making may consider options that vary from delayed treatment including active surveillance (AS) and watchful waiting (WW) to radical prostatectomy (RP) and radiation therapy (RT) with the former being more frequently performed by the robotic approach at high-volume centers [2,3].

Although LR PCa represents the most favorable prognostic category, these patients may have adverse cancer control outcomes due to tumor misclassification related to critical upstaging and/or upgrading at final pathology [4]. According to reports, rates of tumor upstaging to non-organ-confined disease (extra-capsular extension (ECE), seminal vesicle invasion (SVI), or lymph node invasion (LNI)) range from 10.5 to 17.0%, while rates of tumor upgrading range from 43.0 to 63.8%; however, only 1.0% to 6.0% of these patients harbored an International Society of Urological Pathology (ISUP) grade group > 2 [4,5,6,7,8]. Nonetheless, rates of tumor upgrading in low-risk PCa patients have increased in recent years. Tumor misclassification is a potential issue for urologists and radiation oncologists when planning treatment decisions, since upgrading and/or upstaging may represent an indication for active treatments instead of AS or WW [9]. Accordingly, while awaiting the introduction of prognostic molecular markers, standard clinical predictors of disease relapse need to be improved to reduce treatment-related regret, which is becoming an issue in everyday clinical practice [10].

We tested rates of favorable vs. unfavorable tumor upgrading, regarded as the presence of an ISUP grade group equal to 2 vs. >2 at final pathology in a contemporary cohort of LR PCa patients treated with robot-assisted radical prostatectomy (RARP) at a tertiary referral center, in addition to testing the effect of favorable vs. unfavorable tumor upgrading on disease relapse. We hypothesized that levels of tumor grade misclassification might impact differently on PCa relapse. Finally, we investigated clinical and pathological predictors of unfavorable tumor upgrading.

## 2. Materials and Methods

### 2.1. Data Collection, Patient Selection, and Evaluation of Parameters

Approval from the Institutional Review Board of the Azienda Ospedaliera Universitaria Integrata of Verona was obtained, and each patient provided informed consent for data collection and analysis. Data were collected prospectively and retrospectively analyzed. From January 2013 to October 2020, 1143 PCa patients underwent RARP at our institution. Follow-up data were available for 901 patients, of whom 237 were classified as LR according to the EAU classification system in presence of PSA < 10 ng/mL, clinical T stage ≤ 2a, and ISUP grade group 1 [2].

For each patient, age, body mass index (kg/m^2^), preoperative PSA (ng/mL), prostate volume (mL) evaluated by a trans-rectal ultrasound standard ellipsoid method, and PSAD (ng/mL/cc) were collected. Prostate biopsies were performed in an office setting under local anesthesia. After perineal disinfection, biopsy cores were taken transperineally according to a standard template, including 14 cores (6 cores in the peripheral zone and 1 in the transitional zone for each side of the gland). Patients with available preoperative mpMRI showing suspicious areas according to the Prostate Imaging Reporting & Data System (PI-RADS) underwent targeted biopsies (3 to 5 cores for each suspicious area) in addition to systematic ones. Patients with biopsies performed elsewhere were also included in our cohort if the number of cores taken was at least 12, according to the EAU guidelines. The percentage of biopsy-positive cores (BPC), defined as the ratio between positive and total taken cores, was also collected. Clinical staging was assessed by the 2017 version (8th edition) of the Tumor Node Metastasis (TNM) classification system with the clinical T stage only referring to a digital rectal examination. RARP was performed according to the standard transperitoneal approach by experienced surgeons. Extended pelvic lymph node dissection (ePLND) was additionally performed according to guidelines’ recommendations [2,3]. Specifically, the decision was based on clinical factors (pre-biopsy PSA > 7 ng/mL, and percentage of BPC > 20%) indicating an increased risk of tumor upgrading and lymph node invasion (LNI) in the surgical specimen [11,12]. Dissected lymph nodes were submitted in separate packages according to a standard anatomical template including external iliac, internal iliac and obturator, Marcille’s common iliac, and Cloquet’s nodal stations, bilaterally [13,14,15]. Specimens including prostate and eventually dissected lymph nodes were placed into formalin and evaluated by the dedicated pathologist. Prostates were weighted and tumors were graded according to the ISUP grade group system [2,3]. Tumor quantitation was assessed as tumor load (TL) defined as the percentage of prostate involved by cancer; specifically, our dedicated pathologist assessed tumor quantitation by visual estimation of all the glass slides after all microscopically identifiable foci of carcinomas had been circled with a marked pen, according to ISUP association [14]. Surgical margins were stated as positive when cancer invaded the inked surface of the specimen; furthermore, surgical margins were classified as focal and non-focal according to the linear extent [2,3,15,16]. Removed lymph nodes were counted and assessed for cancer invasion. Surgical specimens were staged by the 2017 version (8th edition) of the TNM classification system [2,3].

Although patients were followed up according to guidelines’ recommendations, the decision of further treatment after surgery or at disease relapse was taken in a multidisciplinary setting including urologists, radiation oncologists, and medical oncologists to optimize recommendations with patients’ personal issues [2,3].

### 2.2. Study Design and Outcome of Interest

The purpose of this study was to assess (1) rates of favorable and unfavorable tumor upgrading at radical prostatectomy, regarded as the presence of ISUP grade group at final pathology equal to 2 and >2, respectively, and (2) rate of disease relapse, defined as the unfavorable event occurring for biochemical recurrence and/or local recurrence and/or distant metastasis, in LR PCa patients treated with RARP. Furthermore, predictors, as well as the prognostic impact of tumor upgrading at final pathology on disease relapse, were evaluated.

### 2.3. Statistical Analysis

Descriptive statistics included frequencies and proportions for categorical variables. Medians and interquartile ranges (IQR) were reported for continuously coded variables. The length of time between surgery and disease relapse or the last available follow-up was measured as time to event occurrence. Statistical analyses consisted of two steps. First, univariable and multivariable Cox proportional hazards regression models tested the association between favorable vs. unfavorable tumor upgrading and risk of disease relapse. Eventually, appropriate survival risk curves were generated. Second, the association between clinical and pathological variables and unfavorable tumor upgrading was assessed using univariable and multivariable logistic regression models. Accordingly, hazard ratios (HR) and odds ratios (OR) with a relative 95% confidence interval (CI) were computed. Appropriate Receiver operating characteristic (ROC) curves were generated with the area under the curve (AUC), assessing the discriminative power of the variables. All tests were two-sided with *p* < 0.05 indicating statistical significance. IBM-SPSS version 26.0 (IBM Corp., Released 2019, Armonk, NY, USA) was used for all analyses.

## 3. Results

### 3.1. Descriptive Characteristics of the Study Population

In the overall cohort of 237 surgically treated LR PCa patients, the median age was 65 (IQR 60–69) years and the median PSA was 5.8 (IQR 4.5–7.6) ng/mL (Table 1). Tumor upgrading at final pathology occurred in 158 (66.7%) patients, of whom 98 (41.4% of the entire cohort) had favorable tumor upgrading, and 60 (25.3% of the entire cohort) had unfavorable tumor upgrading. Overall, 18 (7.6%) patients harbored non-organ-confined disease at final pathology, including either ECE or SVI, and 51 (21.5%) had positive surgical margins, which were not focal in 15 cases. Extended pelvic lymph node dissection was performed in 64 (27.0%) patients; the median number of lymph nodes removed was 27 (IQR 20–33) with pelvic LNI only detected in 2 cases. Median follow-up was 53 (IQR 29–69) months. Adjuvant androgen deprivation therapy was given in 20 (8.4%) subjects, and 26 (11.0%) patients received postoperative RT, which was administered in 12 patients with salvage finality. Three patients died due to PCa, of whom two died due to disease relapse.

### 3.2. Prognostic Impact of Favorable vs. Unfavorable Tumor Upgrading on PCa Relapse after Radical Prostatectomy

Overall, disease relapse occurred in 20 (8.4%) patients (Table 1). Patients who experienced disease relapse were more likely to have unfavorable tumor upgrading at radical prostatectomy (50.0% vs. 23.1%), as well as positive surgical margins (50.0% vs. 18.9%). According to univariable Cox regression analysis, either unfavorable tumor upgrading (HR: 5.81, 95% CI: 1.59–21.16, *p* = 0.008) or positive surgical margins (HR: 3.67, 95% CI: 1.51–8.92, *p* = 0.004) were associated with disease relapse. Conversely, favorable tumor upgrading, as well as clinical factors, did not show any statistically significant association with disease relapse (Table 1). Figure 1 also shows the negative prognostic impact of unfavorable tumor upgrading (ISUP grade group > 2) on PCa relapse when compared with subjects with ISUP grade group ≤ 2. According to multivariable Cox regression analysis, unfavorable tumor upgrading remained an independent predictor of disease relapse, even after adjustment for clinical and pathological variables including surgical margins status, TL, and unfavorable tumor stage (Table 2).

### 3.3. Predictors of Unfavorable Tumor Upgrading at Final Pathology

Association of clinical and pathological factors with unfavorable tumor upgrading, which occurred in 60 (25.3%) patients, is reported in Table 3. According to multivariable analysis, age (OR: 1.07, 95% CI: 1.02–1.13, *p* = 0.009) and PSAD ≥ 0.15 ng/mL/cc (OR: 2.32, 95% CI: 1.26–4.27, *p* = 0.007) among clinical factors, TL (OR: 1.03, 95% CI: 1.00–105, *p* = 0.031), and extra-prostatic tumor extension, regarded as the presence of either ECE or SVI (OR: 8.54, 95% CI: 2.88–25.32, *p* < 0.001), among pathological factors, represented independent predictors of unfavorable tumor upgrading. The discrimination power was higher for TL when compared with age and with PSAD, as illustrated in Figure 2.

## 4. Discussion

Low risk is the most favorable PCa prognostic category according to EAU and NCCN classification systems [2,3]. Additionally, a LR category has been also considered by the five-tiered Cambridge Prognostic Groups (CPG) classification, which was developed to improve the risk stratification of non-metastatic PCa at initial diagnosis and to better assess PCa prognosis according to risk level [17]. Accordingly, for the CPG 1 category, which included patients presenting the same inclusion criteria of low-risk EAU and NCCN classifications, 10-year PCa-specific mortality varies from 1.2% for treated PCa patients to 4.2% for untreated PCa patients [17]. In LR PCa patients, adverse cancer control outcomes are mainly related to misclassification issues between clinical and pathological tumor burden, which represents a predictor of disease relapse [2,3,4]. In the low- and intermediate-risk categories, an extensive literature review has recently shown that unfavorable pathology, which involves up to 30% of cases, is an issue for the negative impact on disease relapse that is predicted by short PSA doubling time (PSA-DT) and high tumor grade after surgery [18]. In a large European cohort of PCa patients treated with radical prostatectomy, it has been demonstrated that patients with PSA-DT < 2 years and pathological ISUP grade group > 3, which identifies a subgroup at high risk of biochemical recurrence, are associated with an increased risk of metastatic progression and higher cancer-specific mortality. However, this study included ISUP grade group 3 in the biochemical recurrence LR category; furthermore, the same investigators concluded that the EAU biochemical recurrence groups should be refined in the future [19]. Recently, a large European multicenter study evaluated the impact of different clinical criteria in predicting the risk of unfavorable disease at radical prostatectomy, defined as the presence of ISUP tumor grade group > 2, extra-prostatic extension (ECE or SVI), or LNI in low- and intermediate-risk PCa patients treated with radical prostatectomy. Accordingly, PSA, PSAD, biopsy ISUP grade, and clinical tumor stage resulted in independent predictors of unfavorable pathology, which was detected in 1246 of 6933 (17.9%) cases in the LR category. However, the study suffered from several limitations such as the inclusion criteria and the absence of a central pathology review due to the multicenter nature [9]. Therefore, disease relapse, progression, and mortality may occur even in the LR category in presence of adverse pathology, which is closely related to unfavorable tumor grade associated with ECE and SVI. Our results provide an important contribution to unfolding the issue related to this controversial topic which actively involves both urologists and radiation oncologists. In our study, LR patients who did not experience disease relapse were less likely to bear tumors with unfavorable tumor upgrading, which involved 25.3% of the patient population. In the surgical specimen, LR patients presenting with unfavorable upgrading were more likely to associate with other adverse pathological features including higher TL, adverse tumor stage (ECE or SVI), and positive surgical margins. Clinically, LR patients with pathological ISUP tumor grade ≤ 2 were more likely to be younger and to have PSAD < 0.15 ng/mL/cc. Accordingly, older age and PSAD ≥ 0.15 ng/mL/cc were standard clinical factors stratifying LR patients for unfavorable tumor upgrading, which was a negative prognostic factor of PCa relapse.

Currently, the new ISUP grade group system represents the strongest predictor of PCa biology. It has been validated in 20,845 surgically treated patients, who were evaluated for the occurrence of biochemical recurrence; the HR of grade groups 2 to 5 relative to grade group 1 was 2.2, 7.3, 12.3, and 23.9, respectively, with grade group 2 showing a very good prognosis with rare metastases; unfortunately, the study did not stratify the main endpoint according to the clinical prognostic risk category [2,3,20]. In patients with localized PCa treated with RP, adverse tumor upgrading is the main factor for evaluating adverse pathology, which has drawbacks on disease relapse and progression; however, although adverse pathology indicates negative prognosis, its impact is not the same among risk groups, being more favorable for low-risk classes when compared with the intermediate- and high-risk classes [21]. Recently, a North American study has developed and validated a nomogram predicting adverse pathology, which was defined as the presence of ISUP > 1 and/or ECE/SVI, in low-risk PCa patients treated with surgery; the model, which included standard clinical factors with multiparametric magnetic resonance imaging (mpMRI) findings, showed 87% accuracy. However, it suffered several limitations: being retrospective, including only one high-volume surgeon, and not considering the issue of favorable and unfavorable tumor upgrading, which might have a different prognostic impact [22]. 

The findings of our study are of particular interest for ongoing issues related to the low-risk category. Although tumor upgrading occurred in 66.7% of cases, only 25.3% of patients showed unfavorable tumor upgrading that was associated with older age and PSAD ≥ 0.15 ng/mL/cc and represented a predictor of PCa relapse. Interestingly, ISUP tumor grade group 2, which represented 41.4% of cases, did not have any statistically significant prognostic impact on PCa relapse. These results support EAU and NCCN guidelines recommendations for AS in the favorable intermediate-risk classes [2,3] and might have implications for clinical decision making. In LR PCa patients, when life expectancy is more than ten years, AS is strongly recommended; however, the risk of disease misclassification for tumor undergrading and understaging may become an issue for active treatment being delayed with risk of disease progression [2,3]. Although actual rates of AS for patients with favorable intermediate-risk disease are increasing in North America, more follow-up and research are needed to assess issues related to safety and efficacy [23]. In our study, EAU LR PCa patients were more likely to harbor unfavorable tumor upgrading when presenting older age and having PSAD levels ≥ 0.15 ng/mL/cc; furthermore, patients presenting these features were more likely to experience disease relapse. Accordingly, this information can help either urologists or radiation oncologists when counseling these patients.

Explanations are needed to interpret the results of our study showing that LR PCA patients harboring unfavorable tumor upgrading in the surgical specimen were more likely to be older and to have higher PSAD levels. Theoretically, older patients, compared to their younger counterparts, are more likely to have a longer exposure to genetic mutation dynamics, as well as to have a more compromised immune system, thus unfolding unfavorable tumor grades [24]. Furthermore, the cell population of high-grade cancers may display a high-density growing pattern, eventually enhanced by local stimulating factors. Finally, higher PSAD could be related to more aggressive and extensive tumors growing in smaller prostates. Indeed, PSAD > 0.15 ng/mL/cc increased the probability of detecting clinically significant PCa at biopsy in patients presenting PIRADS 5 lesions at mpMRI [25]. These hypotheses need to be verified by controlled studies.

Our study is not devoid of limitations. First, it is retrospective and single-center. Second, mpMRI, molecular, and/or genetic tests were not evaluated as they were not available in all patients. Third, prostate biopsies performed elsewhere were not reviewed by our dedicated uropathologist. Fourth, we did not evaluate overall mortality-free survival, as well as cancer-specific mortality-free survival due to the limited number of occurred events. Fifth, we did not evaluate the percentage of cancer involving each biopsy core as it was not available in all cases. Nevertheless, our study has strengths. First, disease relapse was considered as the outcome of interest, which represents a stronger endpoint than only earlier biochemical recurrence. Second, all surgical procedures were performed by both low- and high-volume surgeons who did not bias staging results, thus reflecting real-world practice at tertiary referral centers. Third, the length of follow-up was appropriate for evaluating the outcome of interest. Finally, all surgical specimens were evaluated by our dedicated pathologist. 

## 5. Conclusions

In low-risk PCa patients, disease relapse was more likely to occur for unfavorable tumor upgrading (ISUP grade group > 2), which was independently predicted by older age as well as by PSAD ≥ 0.15 ng/mL/cc. Unfavorable tumor upgrading, which associates with larger tumors that eventually extend beyond the prostate, is an issue to consider when counseling LR PCa patients to avoid delayed treatments, which may impair cancer-specific survival.

## Figures and Tables

**Figure 1 cancers-14-06055-f001:**
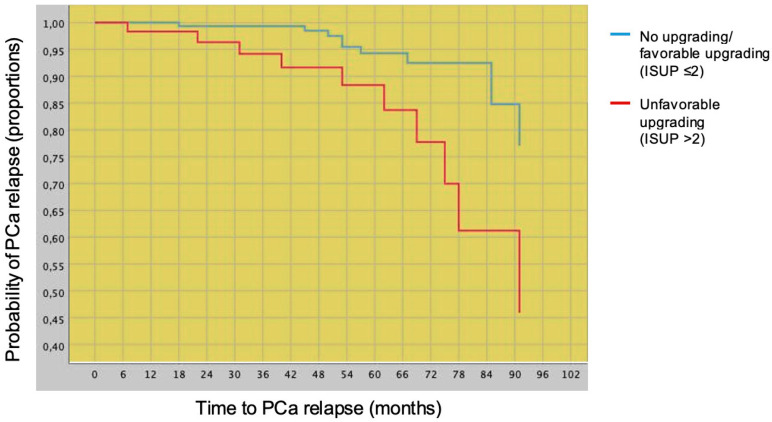
Risk curves depicting time to disease relapse stratified by no tumor upgrading/favorable tumor upgrading (ISUP ≤ 2) vs. unfavorable tumor upgrading (ISUP > 2) in 237 low-risk prostate cancer patients classified according to the European Association of Urology system, treated with robot-assisted radical prostatectomy, with a median follow up of 53 months. On univariable analysis (Cox’s proportional hazards), unfavorable tumor upgrading was a prognostic factor for PCa relapse (hazard ratio, HR: 3.352; 95% CI: 1.393–8.067; *p* = 0.007).

**Figure 2 cancers-14-06055-f002:**
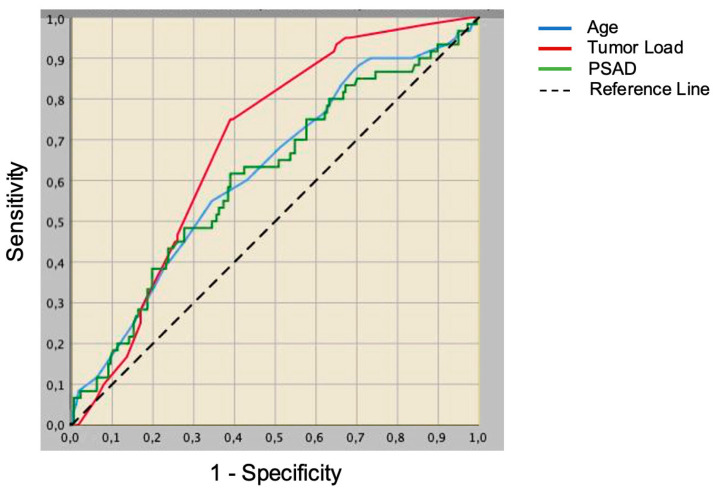
Receiver operating characteristic (ROC) curves of continuous variables predicting unfavorable tumor upgrading in 237 low-risk prostate cancer patients classified according to the European Association of Urology (EAU) system, treated with robot-assisted radical prostatectomy (RARP). Discrimination power was higher for tumor load (area under the curve, AUC = 0.688; 95% CI: 0.618–0.758; *p* < 0.001) when compared with age (AUC = 0.622; 95% CI: 0.541–0.704; *p* = 0.005) and PSA density (AUC = 0.611; 0.527–0.694; *p* = 0.011).

**Table 1 cancers-14-06055-t001:** Descriptive characteristics of 237 low-risk prostate cancer (PCa) patients stratified according to the presence of disease relapse after robot-assisted radical prostatectomy (RARP).

	Overall Cohort *n* = 237 (100%)	No PCa Relapse *n* = 217 (91.6%)	PCa Relapse *n* = 20 (8.4%)	Univariable Analysis	
				HR (95% CI)	*p*-Value
**Clinical factors**					
Age (years)	65 (60–69)	65 (59–69)	66 (61–70)	1.03 (0.96–1.12)	0.411
BMI (kg/m^2^)	26.0 (24.0–28.0)	26.1 (23.9–28.1)	25.5 (24.2–26.8)	0.99 (0.84–1.15)	0.867
PV (mL)	40.0 (30.0–50.0)	40.0 (30.5–50.0)	39.5 (26.2–51.5)	0.99 (0.97–1.02)	0.995
PSA (ng/mL)	5.8 (4.5–7.6)	5.7 (4.5–7.5)	6.5 (4.1–9.0)	1.15 (0.92–1.14)	0.208
PSAD < 0.15 (ng/mL/cc)	128 (54.0)	120 (55.3)	8 (40.0)	*Ref.*	
PSAD ≥ 0.15 (ng/mL/cc)	109 (46.0)	97 (44.7)	12 (60.0)	1.80 (0.74–4.42)	0.198
Biopsy core taken (*n*)	14 (12–15)	14 (12–15)	15 (12–16)		
BPC (%)	25.0 (14.0–35.5)	25.0 (14.0–35.0)	29.0 (16.2–36.0)	1.01 (0.98–1.03)	0.543
cT1	215 (90.7)	197 (90.8)	18 (90.0)	*Ref.*	
cT2a	22 (9.3)	20 (9.2)	2 (10.0)	1.88 (0.43–8.17)	0.401
**Pathological factors**					
ISUP 1	79 (33.3)	76 (35.0)	3 (15.0)	*Ref.*	
ISUP 2	98 (41.4)	91 (41.9)	7 (35.0)	2.52 (0.65–9.76)	0.182
ISUP > 2	60 (25.3)	50 (23.1)	10 (50.0)	5.81 (1.59–21.16)	**0.008**
PW (g)	52.0 (43.0–66.0)	52.0 (43.0–65.5)	54.5 (41.0–72.2)	1.00 (0.98–1.03)	0.949
TL (%)	10.0 (6.2–20.0)	10.0 (5.0–20.0)	15.0 (10.0–20.0)	0.99 (0.95–1.03)	0.749
pT2	219 (92.4)	202 (93.1)	17 (85.0)	*Ref.*	
EPE (ECE or SVI)	18 (7.6)	15 (6.9)	3 (15.0)	2.17 (0.64–7.42)	0.216
Negative surgical margins	186 (78.5)	176 (81.1)	10 (50.0)	*Ref.*	
Positive surgical margins	51 (21.5)	41 (18.9)	10 (50.0)	3.67 (1.51–8.92)	**0.004**

Continuous variables are reported as medians and interquartile ranges, and categorical variables as frequencies and proportions. Abbreviations: HR, hazard ratio; CI, confidence interval; BMI, body mass index; PV, prostate volume; PSA, prostate-specific antigen; PSAD, prostate-specific antigen density; BPC, biopsy-positive cores; ISUP, International Society of Urological Pathology; PW, prostate weight; TL, tumor load; EPE, extra-prostatic extension; ECE, extra-capsular extension; SVI, seminal vesical invasion. Bold fonts indicate statistical significance at *p* < 0.05.

**Table 2 cancers-14-06055-t002:** Multivariable Cox regression models testing the independent predictor status of unfavorable tumor upgrading (ISUP > 2) on disease relapse in 237 low-risk prostate cancer patients treated with robot-assisted radical prostatectomy.

Endpoint	HR (95% CI)	*p*-Value
Hazard ratio adjusted for clinical factors (*)	3.067 (1.168–8.053)	0.023
Hazard ratio adjusted for surgical margins status (*)	2.937 (1.209–7.132)	0.017
Hazard ratio adjusted for tumor load and unfavorable tumor stage (*)	3.935 (1.444–10.723)	0.007

Abbreviations: ISUP, International Society of Urologic Pathology; HR, hazard ratio; CI, confidence interval; (*): see Table 1.

**Table 3 cancers-14-06055-t003:** Logistic regression models testing associations of clinical and pathological factors with unfavorable tumor upgrading in 237 low-risk prostate cancer patients treated with robot-assisted radical prostatectomy.

	No Upgrading (ISUP 1) or Favorable Tumor Upgrading (ISUP 2) *n* = 177 (74.7%)	Unfavorable Tumor Upgrading (ISUP > 2) *n* = 60 (25.3%)	Univariable Analysis		Multivariable Analysis (*)	
			OR (95% CI)	*p*-Value	OR (95% CI)	*p*-Value
**Clinical factors**					**Clinical model**	
Age (years)	65 (58–68)	67 (63–71)	**1.08 (1.02–1.13)**	**0.006**	**1.07 (1.02–1.13)**	**0.009**
BMI (kg/m^2^)	26.2 (24.2–28.0)	25.4 (23.7–27.9)	0.99 (0.90–1.10)	0.857		
PSA (ng/mL)	5.6 (4.5–7.1)	6.5 (4.5–8.7)	**1.18 (1.02–1.36)**	**0.028**		
PV (mL)	40.0 (30.5–52.0)	40.0 (30.0–47.0)	0.99 (0.97–1.01)	0.145		
BPC (%)	24 (14–33)	29 (15.5–42.7)	1.011 (0.994–1.028)	0.202		
PSAD < 0.15 (ng/mL/cc)	105 (59.3)	23 (38.3)	*Ref.*		*Ref.*	
PSAD ≥ 0.15 (ng/mL/cc)	72 (40.7)	37 (61.7)	**2.35 (1.29–4.27)**	**0.005**	**2.32 (1.26–4.27)**	**0.007**
cT1c	165 (93.2)	50 (83.3)	*Ref.*			
cT2a	12 (6.8)	10 (16.7)	**2.75 (1.12–6.74)**	**0.027**		
**Pathological factors**					**Pathological model**	
PW (g)	58.0 (44.0–69.3)	50.5 (42.0–60.0)	0.99 (0.98–1.01)	0.432		
TL (%)	10.0 (5.0–20.0)	15.0 (11.2–24.2)	**1.03 (1.01–1.06)**	**0.011**	**1.03 (1.00–1.05)**	**0.031**
pT2	172 (97.2)	47 (78.3)	*Ref.*		*Ref.*	
EPE (ECE or SVI)	5 (2.8)	13 (21.7)	**9.52 (3.23–28.04)**	**<0.001**	**8.54 (2.88–25.32)**	**<0.001**
Negative surgical margins	148 (83.6)	38 (63.3)	*Ref.*			
Positive surgical margins	29 (16.4)	22 (36.7)	**2.96 (1.60–5.71)**	**<0.001**		

Continuous variables are reported as medians and interquartile ranges, and categorical variables as frequencies and proportions. (*), by the Wald-forward method. Abbreviations: ISUP, International Society of Urological Pathology; OR, odds ratio; CI, confidence interval; BMI, body mass index; PSA, prostate-specific antigen; PV, prostate volume, BPC, biopsy-positive cores; PSAD, prostate-specific antigen density; PW, prostate weight; TL, tumor load; EPE, extra-prostatic extension; ECE, extra-capsular extension; SVI, seminal vesical invasion. Bold fonts indicate statistical significance at *p* < 0.05.

## Data Availability

The data presented in this study are available on request from the corresponding author. The data are not publicly available due to ethical reasons.

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
