# Peer review of "Prognostic Impact and Clinical Implications of Unfavorable Upgrading in Low-Risk Prostate Cancer after Robot-Assisted Radical Prostatectomy: Results of a Single Tertiary Referral Center"

_cancers, 2022, doi:10.3390/cancers14246055_

Round 1

Reviewer 1 Report

This is a single center retrospective study of identifying unfavorable factors in low risk patients undergoing RARP. It is an interesting study of generating hypothesis. My comments and suggestions:

- Abstract: ok

- Introduction: ok

- MM: In the flow-chart there were a lost of follow-up of more than 200 patients. Explain why

- Explain how many (mean, median) lymph nodes were resected

Results:

- The upgrading was 66.7% (more than literature describe, 43-48%). Explain probable reasons

- Describe how many upstaging were found?

- The ADT was related to salvage RT?. otherwise explain why did you use it in LR patients?

- The AUC for tumor load is not stronger¡¡¡ (0.688). Explain why

- In figure 1 introduce median follow-up, HR and CI

Discussion

- Line 201. Begin the sentence with Low-risk (instead of LR)

- Line 286. revise data of PSAd (>15ng/ml(cc?)

The authors have a low PC mortality, so, the clinical impact of these findings are low, even 25% upgrading and 8.4% of relapse. Discuss more about it and make some future recommendations

Author Response

This is a single center retrospective study of identifying unfavorable factors in low-risk patients undergoing RARP. It is an interesting study of generating hypothesis. My comments and suggestions:

- Abstract: ok

- Introduction: ok

- MM: In the flow-chart there were a loss of follow-up of more than 200 patients. Explain why.

We thank the Reviewer for the comment. We retrospectively evaluated data of 1143 patients diagnosed with prostate cancer, who were treated with robot-assisted radical prostatectomy between January 2013 and October 2020 at our Institution. Patients with missing follow-up data (n=242), as well as patients who belong to risk categories other than low-risk (n=664) were excluded from the analysis. These selection criteria yielded an overall cohort of 237 patients. Subjects with unknown/unavailable follow-up data mainly consist of patients coming from other regions of Italy, who were operated at our Institution that represents a referral center for robotic surgical treatment of urological malignancies, and then chose to continue their follow-up near the city of origin. In consequence, detailed information of follow-up of these patients sometimes is missing.

- Explain how many (mean, median) lymph nodes were resected

We thank the Reviewer for the comment. We added the following sentence in the “Results” section of the manuscript.

Extended pelvic lymph node dissection was performed in 64 (27.0%) patients, median lymph nodes removed were 27 (IQR 20-33) with pelvic LNI detected only in 2 cases”.

Results:

- The upgrading was 66.7% (more than literature describe, 43-48%). Explain probable reasons.

We thank the Reviewer for the comment. More than 50% of the overall cohort of low-risk prostate cancer patients treated with robot assisted radical prostatectomy at our Institution underwent prostate biopsies in other centers. Biopsies performed elsewhere were not reviewed by our dedicated uro-pathologist. This factor can probably explain misclassification related issues. However, our rate of tumor upgrading can be compared with other center of excellence, such as Martini-Klinik Prostate Cancer, where ISUP≥2 upgrading at radical prostatectomy was recorded in 63.8% of a cohort of 1,131 patients (Schiffmann J, Wenzel P, Salomon G, et al. Heterogeneity in D'Amico classification-based low-risk prostate cancer: Differences in upgrading and upstaging according to active surveillance eligibility. Urol Oncol. 2015;33(7):329.e13-329.e3.29E19). Conversely, Kovac et al. relied on a larger cohort of low-risk prostate cancer patients treated with radical prostatectomy at Cleveland Clinic Foundation and Memorial Sloan Kettering Cancer Center, and found that of 6371 included patients, 2955 (48%) experienced tumor upgrading at final pathology (Kovac E, Vertosick EA, Sjoberg DD, Vickers AJ, Stephenson AJ. Effects of pathological upstaging or upgrading on metastasis and cancer-specific mortality in men with clinical low-risk prostate cancer. BJU Int. 2018;122(6):1003-1009). These results agree with data coming from studies which relied on two North-American population-based database such as Surveillance, Epidemiology, and End Results, and National Cancer Database, where rates of tumor upgrading ranging from 43 to 46.2% (Caster, J.M.; Falchook, A.D.; Hendrix, L.H.; Chen, R.C. Risk of Pathologic Upgrading or Locally Advanced Disease in Early Prostate Cancer Patients Based on Biopsy Gleason Score and PSA: A Population-Based Study of Modern Patients. Int J Radiat Oncol Biol Phys 2015, 92, 244–251. Flammia, R.S.; Hoeh, B.; Hohenhorst, L.; Sorce, G.; Chierigo, F.; Panunzio, A.; Tian, Z.; Saad, F.; Leonardo, C.; Briganti, A.; et al. Adverse Upgrading and/or Upstaging in Contemporary Low-Risk Prostate Cancer Patients. Int Urol Nephrol 2022, 54, 2521–2528). Nonetheless, Davaro et al. also showed that in low-risk patients rates of upgrading at radical prostatectomy has been increased in recent years, from 41.2 to 56.7% between 2010 and 2016 (Davaro F, Weinstein D, Wong R, Siddiqui S, Hinyard L, Hamilton Z. Increasing rate of pathologic upgrading in low-risk prostate cancer patients in the active surveillance era. Can J Urol. 2022;29(2):11059-11066).

The following sentence in the “Introduction” section of the manuscript was edited:

“According to reports, rates of tumor upstaging to non-organ confined disease (extra-capsular extension [ECE], seminal vesicle invasion [SVI], or lymph node invasion [LNI]) range from 10.5 to 17.1%, while rates of tumor upgrading range from 43.0 to 63.8%; however, only 1.0% to 6.0% of these patients harbored International Society of Urological Pathology (ISUP) grade group > 2. Nonetheless, rates of tumor upgrading in low-risk PCa patients increased in recent years”.

- Describe how many upstaging were found?

We thank the Reviewer for the comment. We recorded 18 (7.6%) tumor upstaging at final pathology, including either extracapsular extension or seminal vesicle invasion. The following sentences was reported:

Overall, 18 (7.6%) patients harbored non-organ confined disease at final pathology, including either ECE or SVI”.

- The ADT was related to salvage RT? otherwise explain why did you use it in LR patients?

We thank the Reviewer for the comment. After treatment patients were followed up according to international guidelines recommendations. The decision to administer further treatments such as androgen-deprivation therapy or radiation therapy has been taken based on final pathology reports (presence of adverse pathology) or at disease relapse (biochemical/local/distant) in a multidisciplinary setting including urologists, radiation oncologists, and medical oncologists of our Institution to optimize guidelines recommendations with patient personal issues.

- The AUC for tumor load is not stronger!!! (0.688). Explain why

We thank the Reviewer for the comment. We agree that the AUC for tumor load can be considered a poor discriminator (<0.7), but in our analysis, it resulted to be the strongest among the other variables considered (age, PSAD).

- In figure 1 introduce median follow-up, HR, and CI

We thank the Reviewer for the comment. As recommended, we added in Figure 1 caption univariable hazard ratio and confidence interval of unfavorable upgrading (ISUP >2) vs. no upgrading/favorable upgrading (ISUP ≤2) in predicting prostate cancer relapse after surgery. Additionally, we added median follow-up, as reported in the manuscript. The new Figure 1 caption is reported below.

“Figure 1. Risk curves depicting time to disease relapse stratified by no tumor upgrading/favorable tumor upgrading (ISUP ≤2) vs. unfavorable tumor upgrading (ISUP >2) in 237 low-risk prostate cancer patients classified according to the European Association of Urology system, treated with robot-assisted radical prostatectomy, with a median follow up of  53 months. On univariable analysis (Cox’s proportional hazards), unfavorable tumor upgrading was a prognostic factor for PCa relapse (hazard ratio, HR: 3.352; 95% CI: 1.393 – 8.067; p = 0.007)”.

Discussion

- Line 201. Begin the sentence with Low-risk (instead of LR)

We thank the Reviewer for the comment. We edited the sentence as recommended.

- Line 286. revise data of PSAd (>15ng/ml(cc?)

We thank the Reviewer for the comment. We apologize for the error. We edited the sentence as follow: “Indeed, PSAD > 0.15 ng/mL/cc increased the probability to detect clinically significant PCa at biopsy in patients presenting PIRADS 5 lesions at mpMRI”.

- The authors have a low PC mortality, so, the clinical impact of these findings are low, even 25% upgrading and 8.4% of relapse. Discuss more about it and make some future recommendations.

We thank the Reviewer for the comment. We agree with the Reviewer about this consideration. However, our endpoint of interest was not cancer-specific mortality, which itself is low for localized prostate cancer and even more so in low-risk, according to previous reports (Kovac E, Vertosick EA, Sjoberg DD, Vickers AJ, Stephenson AJ. Effects of pathological upstaging or upgrading on metastasis and cancer-specific mortality in men with clinical low-risk prostate cancer. BJU Int. 2018;122(6):1003-1009. Boorjian SA, Karnes RJ, Rangel LJ, Bergstralh EJ, Blute ML. Mayo Clinic validation of the D'amico risk group classification for predicting survival following radical prostatectomy. J Urol. 2008;179(4):1354-1361), and contemporary international guidelines data. Instead, we focused on impact of tumor upgrading on disease relapse in low-risk PCa patients.

We hope that the Reviewer will find our edits satisfactory.

Reviewer 2 Report

The paper demonstrates the value of ISUP upgrading after RARP for low-risk prostate cancer. The paper is well written, methodology is correct, results are clearly presented. But, some changes and explanations are required.

Please see the details in the following points.

1.      How was the prostate biopsy performed? Was it systematic and targeted one? How many cores were sampled? This is the main factor responsible for under-grading at initial diagnosis. Was the MRI used before the biopsies? How many cores were sampled on average? All information about the biopsy should be added to the methods and the biopsy results should be added to the table 1. Comments in the discussion regarding the biopsy and related bias are required.

2.      How would you comment very high prevalence of upgrading in your cohort (66%)? It is more than in other studies.

3.      “LR PCa patients who harbored unfavorable tumor upgrading at radical prostatectomy (ISUP grade group > 2) were more likely to experience disease relapse, which occurred in 8.4% of cases and was associated with older age as well as with PSAD ≥ 0.15 304 ng/mL/cc.”

This sentence from the conclusion requires clarification. It sounds like the relapse is associated with older age and higher PSAD, but you do not show this in multivariable analysis. These factors were associated only with upgrading. On the other hand, PSM which is associated with relapse was not mentioned. Please correct accordingly.

Author Response

The paper demonstrates the value of ISUP upgrading after RARP for low-risk prostate cancer. The paper is well written, methodology is correct, results are clearly presented. But some changes and explanations are required. Please see the details in the following points.

We thank the Reviewer for the comments. We really appreciate it.

  1. How was the prostate biopsy performed? Was it systematic and targeted one? How many cores were sampled? This is the main factor responsible for under-grading at initial diagnosis. Was the MRI used before the biopsies? How many cores were sampled on average? All information about the biopsy should be added to the methods and the biopsy results should be added to the table 1. Comments in the discussion regarding the biopsy and related bias are required.

We thank the Reviewer for the comments. The following passage regarding how prostate biopsies are performed at our Institution was added.

 “Prostate biopsies were performed in an office setting under local anesthesia. After perineal disinfection, biopsy cores were taken transperineally according to a standard template, including 14 cores (6 cores in the peripheral zone and 1 in the transitional zone for each side of the gland). Patients with available preoperative mpMRI showing suspicious areas according to Prostate Imaging Reporting & Data System (PI-RADS) underwent targeted biopsies (3 to 5 cores for each suspicious area) in addition to systematic ones. Patients with biopsies performed elsewhere were also included in our cohort if the number of cores taken was at least 12, according to the EAU guidelines”.

We also added the biopsy results to Table 1. We considered only the BPC for the statistical analysis, according to the EAU guidelines.

  1. How would you comment very high prevalence of upgrading in your cohort (66%)? It is more than in other studies.

We thank the Reviewer for the comment. More than 50% of the overall cohort of low-risk prostate cancer patients treated with robot assisted radical prostatectomy at our Institution underwent prostate biopsies in other centers. Unfortunately, biopsies performed elsewhere were not reviewed by our dedicated uro-pathologist. This factor can probably explain misclassification related issues, and was added in the limitation’s’ section of the manuscript. However, our rate of tumor upgrading can be compared with other center of excellence, such as Martini-Klinik Prostate Cancer, where ISUP≥2 upgrading at radical prostatectomy was recorded in 63.8% of a cohort of 1,131 patients (Schiffmann J, Wenzel P, Salomon G, et al. Heterogeneity in D'Amico classification-based low-risk prostate cancer: Differences in upgrading and upstaging according to active surveillance eligibility. Urol Oncol. 2015;33(7):329.e13-329.e3.29E19). Conversely, Kovac et al. relied on a larger cohort of low-risk prostate cancer patients treated with radical prostatectomy at Cleveland Clinic Foundation and Memorial Sloan Kettering Cancer Center, and found that of 6371 included patients, 2955 (48%) experienced tumor upgrading at final pathology (Kovac E, Vertosick EA, Sjoberg DD, Vickers AJ, Stephenson AJ. Effects of pathological upstaging or upgrading on metastasis and cancer-specific mortality in men with clinical low-risk prostate cancer. BJU Int. 2018;122(6):1003-1009). These results agree with data coming from studies which relied on two North-American population-based database such as Surveillance, Epidemiology, and End Results, and National Cancer Database, where rates of tumor upgrading ranging from 43 to 46.2% (Caster, J.M.; Falchook, A.D.; Hendrix, L.H.; Chen, R.C. Risk of Pathologic Upgrading or Locally Advanced Disease in Early Prostate Cancer Patients Based on Biopsy Gleason Score and PSA: A Population-Based Study of Modern Patients. Int J Radiat Oncol Biol Phys 2015, 92, 244–251. Flammia, R.S.; Hoeh, B.; Hohenhorst, L.; Sorce, G.; Chierigo, F.; Panunzio, A.; Tian, Z.; Saad, F.; Leonardo, C.; Briganti, A.; et al. Adverse Upgrading and/or Upstaging in Contemporary Low-Risk Prostate Cancer Patients. Int Urol Nephrol 2022, 54, 2521–2528). Nonetheless, Davaro et al. also showed that in low-risk patients rates of upgrading at radical prostatectomy has been increased in recent years, from 41.2 to 56.7% between 2010 and 2016 (Davaro F, Weinstein D, Wong R, Siddiqui S, Hinyard L, Hamilton Z. Increasing rate of pathologic upgrading in low-risk prostate cancer patients in the active surveillance era. Can J Urol. 2022;29(2):11059-11066).

The following sentence in the “Introduction” section of the manuscript was edited:

“According to reports, rates of tumor upstaging to non-organ confined disease (extra-capsular extension [ECE], seminal vesicle invasion [SVI], or lymph node invasion [LNI]) range from 10.5 to 17.1%, while rates of tumor upgrading range from 43.0 to 63.8%; however, only 1.0% to 6.0% of these patients harbored International Society of Urological Pathology (ISUP) grade group > 2. Nonetheless, rates of tumor upgrading in low-risk PCa patients increased in recent years”.

  1. “LR PCa patients who harbored unfavorable tumor upgrading at radical prostatectomy (ISUP grade group > 2) were more likely to experience disease relapse, which occurred in 8.4% of cases and was associated with older age as well as with PSAD ≥ 0.15 ng/mL/cc.” This sentence from the conclusion requires clarification. It sounds like the relapse is associated with older age and higher PSAD, but you do not show this in multivariable analysis. These factors were associated only with upgrading. On the other hand, PSM which is associated with relapse was not mentioned. Please correct accordingly.

We thank the Reviewer for the comment. We apologized for the mistake. We revised the conclusions as follow: “In low-risk PCa patients, disease relapse was more likely to occur for unfavorable tumor upgrading (ISUP grade group > 2), which was independently predicted by older age as well as by PSAD ≥ 0.15 ng/mL/cc”.

Also, PSM was not reported because it didn’t contribute to predicting the unfavorable tumor upgrading in multivariable analysis.

 We hope that the Reviewer will find our edits satisfactory.

Round 2

Reviewer 2 Report

Thank You